# Zoneout: Regularizing RNNs by Randomly Preserving Hidden Activations

**David Krueger**[1,*], **Tegan Maharaj**[2,*], **János Kramár**[2]
**Mohammad Pezeshki**[1] **Nicolas Ballas**[1], **Nan Rosemary Ke**[2], **Anirudh Goyal**[1]
**Yoshua Bengio**[1†], **Aaron Courville**[1‡], **Christopher Pal**[2]
[1] MILA, Université de Montréal, `firstname.lastname@umontreal.ca`.
[2] École Polytechnique de Montréal, `firstname.lastname@polymtl.ca`.
[*] Equal contributions. [†]CIFAR Senior Fellow. [‡]CIFAR Fellow.

## Abstract

We propose zoneout, a novel method for regularizing RNNs. At each timestep, zoneout stochastically forces some hidden units to maintain their previous values. Like dropout, zoneout uses random noise to train a pseudo-ensemble, improving generalization. But by preserving instead of dropping hidden units, gradient information and state information are more readily propagated through time, as in feedforward stochastic depth networks. We perform an empirical investigation of various RNN regularizers, and find that zoneout gives significant performance improvements across tasks. We achieve competitive results with relatively simple models in character- and word-level language modelling on the Penn Treebank and Text8 datasets, and combining with recurrent batch normalization (Cooijmans et al., 2016) yields state-of-the-art results on permuted sequential MNIST.

## 1 Introduction

Regularizing neural nets can significantly improve performance, as indicated by the widespread use of early stopping, and success of regularization methods such as dropout and its recurrent variants (Hinton et al., 2012; Srivastava et al., 2014; Zaremba et al., 2014; Gal, 2015). In this paper, we address the issue of regularization in recurrent neural networks (RNNs) with a novel method called **zoneout**.

RNNs sequentially construct fixed-length representations of arbitrary-length sequences by folding new observations into their hidden state using an input-dependent transition operator. The repeated application of the same transition operator at the different time steps of the sequence, however, can make the dynamics of an RNN sensitive to minor perturbations in the hidden state; the transition dynamics can magnify components of these perturbations exponentially. Zoneout aims to improve RNNs' robustness to perturbations in the hidden state in order to regularize transition dynamics.

Like dropout, zoneout injects noise during training. But instead of setting some units' activations to 0 as in dropout, zoneout randomly replaces some units' activations with their activations from the previous timestep. As in dropout, we use the expectation of the random noise at test time. This results in a simple regularization approach which can be applied through time for any RNN architecture, and can be conceptually extended to any model whose state varies over time.

Compared with dropout, zoneout is appealing because it preserves information flow forwards and backwards through the network. This helps combat the vanishing gradient problem (Hochreiter, 1991; Bengio et al., 1994), as we observe experimentally.

We also empirically evaluate zoneout on classification using the permuted sequential MNIST dataset, and on language modelling using the Penn Treebank and Text8 datasets, demonstrating competitive or state of the art performance across tasks. In particular, we show that zoneout performs competitively with other proposed regularization methods for RNNs, including recently-proposed dropout variants. Code for replicating all experiments can be found at: `http://github.com/teganmaharaj/zoneout`

## 2 RELATED WORK

### 2.1 RELATIONSHIP TO DROPOUT

Zoneout can be seen as a selective application of dropout to some of the nodes in a modified computational graph, as shown in Figure 1. In zoneout, instead of dropping out (being set to 0), units *zone out* and are set to their previous value ($h_t = h_{t-1}$). Zoneout, like dropout, can be viewed as a way to train a pseudo-ensemble (Bachman et al., 2014), injecting noise using a stochastic "identity-mask" rather than a zero-mask. We conjecture that identity-masking is more appropriate for RNNs, since it makes it easier for the network to preserve information from previous timesteps going forward, and facilitates, rather than hinders, the flow of gradient information going backward, as we demonstrate experimentally.

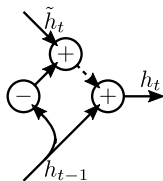

Figure 1: Zoneout as a special case of dropout; $\tilde{h}_t$ is the unit $h$'s hidden activation for the next time step (if not zoned out). Zoneout can be seen as applying dropout on the hidden state delta, $\tilde{h}_t - h_{t-1}$. When this update is dropped out (represented by the dashed line), $h_t$ becomes $h_{t-1}$.

### 2.2 DROPOUT IN RNNS

Initially successful applications of dropout in RNNs (Pham et al., 2013; Zaremba et al., 2014) only applied dropout to feed-forward connections ("up the stack"), and not recurrent connections ("forward through time"), but several recent works (Semeniuta et al., 2016; Moon et al., 2015; Gal, 2015) propose methods that are not limited in this way. Bayer et al. (2013) successfully apply fast dropout (Wang & Manning, 2013), a deterministic approximation of dropout, to RNNs.

Semeniuta et al. (2016) apply **recurrent dropout** to the *updates* to LSTM memory cells (or GRU states), i.e. they drop out the input/update gate in LSTM/GRU. Like zoneout, their approach prevents the loss of long-term memories built up in the states/cells of GRUs/LSTMS, but zoneout does this by preserving units' activations *exactly*. This difference is most salient when zoning out the hidden states (not the memory cells) of an LSTM, for which there is no analogue in recurrent dropout. Whereas saturated output gates or output nonlinearities would cause recurrent dropout to suffer from vanishing gradients (Bengio et al., 1994), zoned-out units still propagate gradients effectively in this situation. Furthermore, while the recurrent dropout method is specific to LSTMs and GRUs, zoneout generalizes to any model that sequentially builds distributed representations of its input, including vanilla RNNs.

Also motivated by preventing memory loss, Moon et al. (2015) propose **rnnDrop**. This technique amounts to using the same dropout mask at every timestep, which the authors show results in improved performance on speech recognition in their experiments. Semeniuta et al. (2016) show, however, that past states' influence vanishes exponentially as a function of dropout probability when taking the expectation at test time in rnnDrop; this is problematic for tasks involving longer-term dependencies.

Gal (2015) propose another technique which uses the same mask at each timestep. Motivated by variational inference, they drop out the rows of weight matrices in the input and output embeddings and LSTM gates, instead of dropping units' activations. The proposed **variational RNN** technique achieves single-model state-of-the-art test perplexity of 73.4 on word-level language modelling of Penn Treebank.

### 2.3 RELATIONSHIP TO STOCHASTIC DEPTH

Zoneout can also be viewed as a per-unit version of **stochastic depth** (Huang et al., 2016), which randomly drops entire layers of feed-forward residual networks (ResNets (He et al., 2015)). This is

equivalent to zoning out all of the units of a layer at the same time. In a typical RNN, there is a new input at each timestep, causing issues for a naive implementation of stochastic depth. Zoning out an entire layer in an RNN means the input at the corresponding timestep is completely ignored, whereas zoning out individual units allows the RNN to take each element of its input sequence into account. We also found that using residual connections in recurrent nets led to instability, presumably due to the parameter sharing in RNNs. Concurrent with our work, Singh et al. (2016) propose zoneout for ResNets, calling it **SkipForward**. In their experiments, zoneout is outperformed by stochastic depth, dropout, and their proposed **Swapout** technique, which randomly drops either or both of the identity or residual connections. Unlike Singh et al. (2016), we apply zoneout to RNNs, and find it outperforms stochastic depth and recurrent dropout.

## 2.4    SELECTIVELY UPDATING HIDDEN UNITS

Like zoneout, **clockwork RNNs** (Koutnik et al., 2014) and **hierarchical RNNs** (Hihi & Bengio, 1996) update only some units' activations at every timestep, but their updates are periodic, whereas zoneout's are stochastic. Inspired by clockwork RNNs, we experimented with zoneout variants that target different update rates or schedules for different units, but did not find any performance benefit. **Hierarchical multiscale LSTMs** (Chung et al., 2016) learn update probabilities for different units using the straight-through estimator (Bengio et al., 2013; Courbariaux et al., 2015), and combined with recently-proposed Layer Normalization (Ba et al., 2016), achieve competitive results on a variety of tasks. As the authors note, their method can be interpreted as an input-dependent form of adaptive zoneout.

In recent work, Ha et al. (2016) use a hypernetwork to dynamically rescale the row-weights of a primary LSTM network, achieving state-of-the-art 1.21 BPC on character-level Penn Treebank when combined with layer normalization (Ba et al., 2016) in a two-layer network. This scaling can be viewed as an adaptive, differentiable version of the variational LSTM (Gal, 2015), and could similarly be used to create an adaptive, differentiable version of zoneout. Very recent work conditions zoneout probabilities on suprisal (a measure of the discrepancy between the predicted and actual state), and sets a new state of the art on enwik8 (Rocki et al., 2016).

## 3    ZONEOUT AND PRELIMINARIES

We now explain zoneout in full detail, and compare with other forms of dropout in RNNs. We start by reviewing recurrent neural networks (RNNs).

### 3.1    RECURRENT NEURAL NETWORKS

Recurrent neural networks process data $x_1, x_2, \ldots, x_T$ sequentially, constructing a corresponding sequence of representations, $h_1, h_2, \ldots, h_T$. Each hidden state is trained (implicitly) to remember and emphasize all task-relevant aspects of the preceding inputs, and to incorporate new inputs via a transition operator, $\mathcal{T}$, which converts the present hidden state and input into a new hidden state: $h_t = \mathcal{T}(h_{t-1}, x_t)$. Zoneout modifies these dynamics by mixing the original transition operator $\tilde{\mathcal{T}}$ with the identity operator (as opposed to the null operator used in dropout), according to a vector of Bernoulli masks, $d_t$:

$$\text{Zoneout:} \quad \mathcal{T} = d_t \odot \tilde{\mathcal{T}} + (1 - d_t) \odot 1 \qquad \text{Dropout:} \quad \mathcal{T} = d_t \odot \tilde{\mathcal{T}} + (1 - d_t) \odot 0$$

### 3.2    LONG SHORT-TERM MEMORY

In long short-term memory RNNs (LSTMs) (Hochreiter & Schmidhuber, 1997), the hidden state is divided into memory cell $c_t$, intended for internal long-term storage, and hidden state $h_t$, used as a transient representation of state at timestep $t$. In the most widely used formulation of an LSTM (Gers et al., 2000), $c_t$ and $h_t$ are computed via a set of four "gates", including the forget gate, $f_t$, which directly connects $c_t$ to the memories of the previous timestep $c_{t-1}$, via an element-wise multiplication. Large values of the forget gate cause the cell to remember most (not all) of its previous value. The other gates control the flow of information in ($i_t, g_t$) and out ($o_t$) of the cell. Each gate has a weight matrix and bias vector; for example the forget gate has $W_{xf}$, $W_{hf}$, and $b_f$. For brevity, we will write these as $W_x, W_h, b$.

An LSTM is defined as follows:

$$i_t, f_t, o_t = \sigma(W_x x_t + W_h h_{t-1} + b)$$
$$g_t = \tanh(W_{xg} x_t + W_{hg} h_{t-1} + b_g)$$
$$c_t = f_t \odot c_{t-1} + i_t \odot g_t$$
$$h_t = o_t \odot \tanh(c_t)$$

A naive application of dropout in LSTMs would zero-mask either or both of the memory cells and hidden states, without changing the computation of the gates $(i, f, o, g)$. Dropping memory cells, for example, changes the computation of $c_t$ as follows:

$$c_t = d_t \odot (f_t \odot c_{t-1} + i_t \odot g_t)$$

Alternatives abound, however; masks can be applied to any subset of the gates, cells, and states. Semeniuta et al. (2016), for instance, zero-mask the input gate:

$$c_t = (f_t \odot c_{t-1} + d_t \odot i_t \odot g_t)$$

When the input gate is masked like this, there is no additive contribution from the input or hidden state, and the value of the memory cell simply decays according to the forget gate.

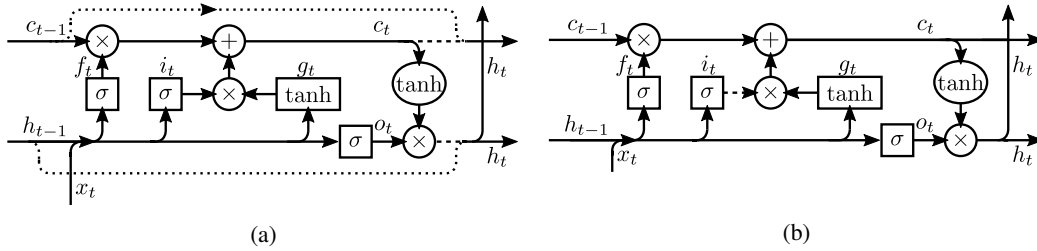

(a)  (b)

Figure 2: (a) Zoneout, vs (b) the recurrent dropout strategy of (Semeniuta et al., 2016) in an LSTM. Dashed lines are zero-masked; in zoneout, the corresponding dotted lines are masked with the corresponding opposite zero-mask. Rectangular nodes are embedding layers.

In **zoneout**, the values of the hidden state and memory cell randomly either maintain their previous value or are updated as usual. This introduces stochastic identity connections between subsequent time steps:

$$c_t = d_t^c \odot c_{t-1} + (1 - d_t^c) \odot \left( f_t \odot c_{t-1} + i_t \odot g_t \right)$$
$$h_t = d_t^h \odot h_{t-1} + (1 - d_t^h) \odot \left( o_t \odot \tanh \left( f_t \odot c_{t-1} + i_t \odot g_t \right) \right)$$

We usually use different zoneout masks for cells and hiddens. We also experiment with a variant of recurrent dropout that reuses the input dropout mask to zoneout the corresponding output gates:

$$c_t = (f_t \odot c_{t-1} + d_t \odot i_t \odot g_t)$$
$$h_t = ((1 - d_t) \odot o_t + d_t \odot o_{t-1}) \odot \tanh(c_t)$$

The motivation for this variant is to prevent the network from being forced (by the output gate) to expose a memory cell which has not been updated, and hence may contain misleading information.

## 4 EXPERIMENTS AND DISCUSSION

We evaluate zoneout's performance on the following tasks: (1) Character-level language modelling on the Penn Treebank corpus (Marcus et al., 1993); (2) Word-level language modelling on the Penn Treebank corpus (Marcus et al., 1993); (3) Character-level language modelling on the Text8 corpus (Mahoney, 2011); (4) Classification of hand-written digits on permuted sequential MNIST (pMNIST) (Le et al., 2015). We also investigate the gradient flow to past hidden states, using pMNIST.

## 4.1 PENN TREEBANK LANGUAGE MODELLING DATASET

The Penn Treebank language model corpus contains 1 million words. The model is trained to predict the next word (evaluated on perplexity) or character (evaluated on BPC: bits per character) in a sequence. [1]

### 4.1.1 CHARACTER-LEVEL

For the character-level task, we train networks with one layer of 1000 hidden units. We train LSTMs with a learning rate of 0.002 on overlapping sequences of 100 in batches of 32, optimize using Adam, and clip gradients with threshold 1. These settings match those used in Cooijmans et al. (2016). We also train GRUs and tanh-RNNs with the same parameters as above, except sequences are non-overlapping and we use learning rates of 0.001, and 0.0003 for GRUs and tanh-RNNs respectively. Small values (0.1, 0.05) of zoneout significantly improve generalization performance for all three models. Intriguingly, we find zoneout increases training time for GRU and tanh-RNN, but *decreases* training time for LSTMs.

We focus our investigation on LSTM units, where the dynamics of zoning out states, cells, or both provide interesting insight into zoneout's behaviour. Figure 3 shows our exploration of zoneout in LSTMs, for various zoneout probabilities of cells and/or hiddens. Zoneout on cells with probability 0.5 or zoneout on states with probability 0.05 both outperform the best-performing recurrent dropout ($p = 0.25$). Combining $z_c = 0.5$ and $z_h = 0.05$ leads to our best-performing model, which achieves 1.27 BPC, competitive with recent state-of-the-art set by (Ha et al., 2016). We compare zoneout to recurrent dropout (for $p \in \{0.05, 0.2, 0.25, 0.5, 0.7\}$), weight noise ($\sigma = 0.075$), norm stabilizer ($\beta = 50$) (Krueger & Memisevic, 2015), and explore stochastic depth (Huang et al., 2016) in a recurrent setting (analogous to zoning out an entire timestep). We also tried a shared-mask variant of zoneout as used in $p$MNIST experiments, where the same mask is used for both cells and hiddens. Neither stochastic depth or shared-mask zoneout performed as well as separate masks, sampled per unit. Figure 3 shows the best performance achieved with each regularizer, as well as an unregularized LSTM baseline. Results are reported in Table 1, and learning curves shown in Figure 4.

Low zoneout probabilities (0.05-0.25) also improve over baseline in GRUs and tanh-RNNs, reducing BPC from 1.53 to 1.41 for GRU and 1.67 to 1.52 for tanh-RNN. Similarly, low zoneout probabilities work best on the hidden states of LSTMs. For memory cells in LSTMs, however, higher probabilities (around 0.5) work well, perhaps because large forget-gate values approximate the effect of cells zoning out. We conjecture that best performance is achieved with zoneout LSTMs because of the stability of having both state and cell. The probability that both will be zoned out is very low, but having one or the other zoned out carries information from the previous timestep forward, while having the other react 'normally' to new information.

### 4.1.2 WORD-LEVEL

For the word-level task, we replicate settings from Zaremba et al. (2014)'s best single-model performance. This network has 2 layers of 1500 units, with weights initialized uniformly [-0.04, +0.04]. The model is trained for 14 epochs with learning rate 1, after which the learning rate is reduced by a factor of 1.15 after each epoch. Gradient norms are clipped at 10.

With no dropout on the non-recurrent connections (i.e. zoneout as the only regularization), we do not achieve competitive results. We did not perform any search over models, and conjecture that the large model size requires regularization of the feed-forward connections. Adding zoneout ($z_c = 0.25$ and $z_h = 0.025$) on the recurrent connections to the model optimized for dropout on the non-recurrent connections however, we are able to improve test perplexity from 78.4 to 77.4. We report the best performance achieved with a given technique in Table 1.

---

[1] These metrics are deterministic functions of negative log-likelihood (NLL). Specifically, perplexity is exponentiated NLL, and BPC (entropy) is NLL divided by the natural logarithm of 2.

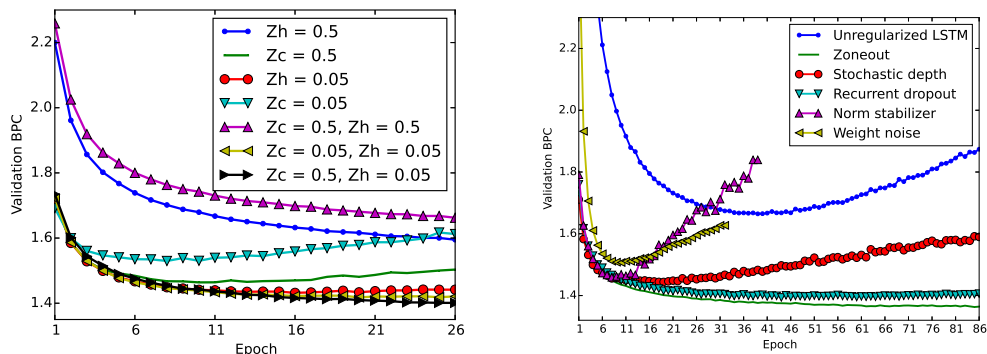

Figure 3: Validation BPC (bits per character) on Character-level Penn Treebank, for different probabilities of zoneout on cells $z_c$ and hidden states $z_h$ (left), and comparison of an unregularized LSTM, zoneout $z_c = 0.5, z_h = 0.05$, stochastic depth zoneout $z = 0.05$, recurrent dropout $p = 0.25$, norm stabilizer $\beta = 50$, and weight noise $\sigma = 0.075$ (right).

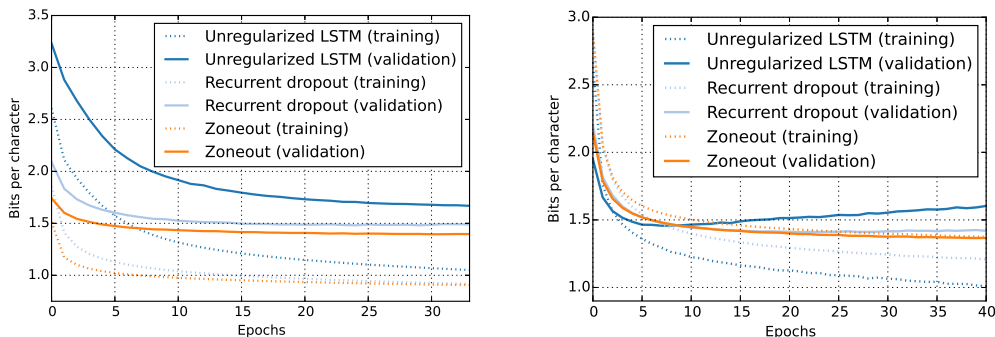

Figure 4: Training and validation bits-per-character (BPC) comparing LSTM regularization methods on character-level Penn Treebank (left) and Text8. (right)

## 4.2 TEXT8

Enwik8 is a corpus made from the first $10^9$ bytes of Wikipedia dumped on Mar. 3, 2006. Text8 is a "clean text" version of this corpus; with html tags removed, numbers spelled out, symbols converted to spaces, all lower-cased. Both datasets were created and are hosted by Mahoney (2011).

We use a single-layer network of 2000 units, initialized orthogonally, with batch size 128, learning rate 0.001, and sequence length 180. We optimize with Adam (Kingma & Ba, 2014), clip gradients to a maximum norm of 1 (Pascanu et al., 2012), and use early stopping, again matching the settings of Cooijmans et al. (2016). Results are reported in Table 1, and Figure 4 shows training and validation learning curves for zoneout ($z_c = 0.5, z_h = 0.05$) compared to an unregularized LSTM and to recurrent dropout.

## 4.3 PERMUTED SEQUENTIAL MNIST

In sequential MNIST, pixels of an image representing a number [0-9] are presented one at a time, left to right, top to bottom. The task is to classify the number shown in the image. In $p$MNIST , the pixels are presented in a (fixed) random order.

We compare recurrent dropout and zoneout to an unregularized LSTM baseline. All models have a single layer of 100 units, and are trained for 150 epochs using RMSProp (Tieleman & Hinton, 2012) with a decay rate of 0.5 for the moving average of gradient norms. The learning rate is set to 0.001 and the gradients are clipped to a maximum norm of 1 (Pascanu et al., 2012).

As shown in Figure 5 and Table 2, zoneout gives a significant performance boost compared to the LSTM baseline and outperforms recurrent dropout (Semeniuta et al., 2016), although recurrent batch normalization (Cooijmans et al., 2016) outperforms all three. However, by adding zoneout to the recurrent batch normalized LSTM, we achieve state of the art performance. For this setting, the zoneout mask is shared between cells and states, and the recurrent dropout probability and zoneout probabilities are both set to 0.15.

Table 1: Validation and test results of different models on the three language modelling tasks. Results are reported for the best-performing settings. Performance on Char-PTB and Text8 is measured in bits-per-character (BPC); Word-PTB is measured in perplexity. For Char-PTB and Text8 all models are 1-layer unless otherwise noted; for Word-PTB all models are 2-layer. Results above the line are from our own implementation and experiments. Models below the line are: NR-dropout (non-recurrent dropout), V-Dropout (variational dropout), RBN (recurrent batchnorm), H-LSTM+LN (HyperLSTM + LayerNorm), 3-HM-LSTM+LN (3-layer Hierarchical Multiscale LSTM + LayerNorm).

| | Char-PTB | | Word-PTB | | Text8 | |
|---|---|---|---|---|---|---|
| Model | Valid | Test | Valid | Test | Valid | Test |
| Unregularized LSTM | 1.466 | 1.356 | 120.7 | 114.5 | 1.396 | 1.408 |
| Weight noise | 1.507 | 1.344 | – | – | 1.356 | 1.367 |
| Norm stabilizer | 1.459 | 1.352 | – | – | 1.382 | 1.398 |
| Stochastic depth | 1.432 | 1.343 | – | – | 1.337 | 1.343 |
| Recurrent dropout | 1.396 | 1.286 | 91.6 | 87.0 | 1.386 | 1.401 |
| Zoneout | 1.362 | 1.252 | 81.4 | 77.4 | 1.331 | 1.336 |
| NR-dropout (Zaremba et al., 2014) | – | – | 82.2 | 78.4 | – | – |
| V-dropout (Gal, 2015) | – | – | – | **73.4** | – | – |
| RBN (Cooijmans et al., 2016) | – | 1.32 | – | – | – | 1.36 |
| H-LSTM + LN (Ha et al., 2016) | 1.281 | 1.250 | – | – | – | – |
| 3-HM-LSTM + LN (Chung et al., 2016) | – | **1.24** | – | – | – | **1.29** |

Table 2: Error rates on the pMNIST digit classification task. Zoneout outperforms recurrent dropout, and sets state of the art when combined with recurrent batch normalization.

| Model | Valid | Test |
|---|---|---|
| Unregularized LSTM | 0.092 | 0.102 |
| Recurrent dropout $p = 0.5$ | 0.083 | 0.075 |
| Zoneout $z_c = z_h = 0.15$ | 0.063 | 0.069 |
| Recurrent batchnorm | - | 0.046 |
| Recurrent batchnorm & Zoneout $z_c = z_h = 0.15$ | 0.045 | **0.041** |

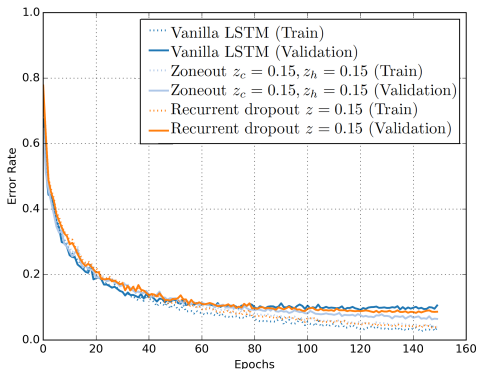

Figure 5: Training and validation error rates for an unregularized LSTM, recurrent dropout, and zoneout on the task of permuted sequential MNIST digit classification.

## 4.4 Gradient flow

We investigate the hypothesis that identity connections introduced by zoneout facilitate gradient flow to earlier timesteps. Vanishing gradients are a perennial issue in RNNs. As effective as many techniques are for mitigating vanishing gradients (notably the LSTM architecture Hochreiter & Schmidhuber (1997)), we can always imagine a longer sequence to train on, or a longer-term dependence we want to capture.

We compare gradient flow in an unregularized LSTM to zoning out (stochastic identity-mapping) and dropping out (stochastic zero-mapping) the recurrent connections after one epoch of training on $p$MNIST. We compute the average gradient norms $\|\frac{\partial L}{\partial c_t}\|$ of loss $L$ with respect to cell activations $c_t$ at each timestep $t$, and for each method, normalize the average gradient norms by the sum of average gradient norms for all timesteps.

Figure 6 shows that zoneout propagates gradient information to early timesteps much more effectively than dropout on the recurrent connections, and even more effectively than an unregularized LSTM. The same effect was observed for hidden states $h_t$.

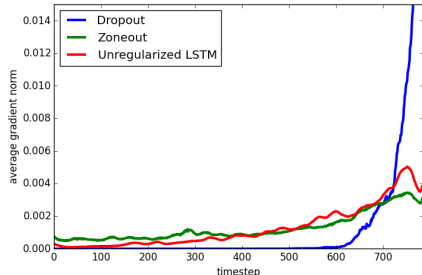

Figure 6: Normalized $\sum \|\frac{\partial L}{\partial c_t}\|$ of loss $L$ with respect to cell activations $c_t$ at each timestep $t$ for zoneout ($z_c = 0.5$), dropout ($z_c = 0.5$), and an unregularized LSTM on one epoch of $p$MNIST
.

## 5 Conclusion

We have introduced zoneout, a novel and simple regularizer for RNNs, which stochastically preserves hidden units' activations. Zoneout improves performance across tasks, outperforming many alternative regularizers to achieve results competitive with state of the art on the Penn Treebank and Text8 datasets, and state of the art results on $p$MNIST. While searching over zoneout probabilites allows us to tune zoneout to each task, low zoneout probabilities (0.05 - 0.2) on states reliably improve performance of existing models.

We perform no hyperparameter search to achieve these results, simply using settings from the previous state of the art. Results on $p$MNIST and word-level Penn Treebank suggest that Zoneout works well in combination with other regularizers, such as recurrent batch normalization, and dropout on feedforward/embedding layers. We conjecture that the benefits of zoneout arise from two main factors: (1) Introducing stochasticity makes the network more robust to changes in the hidden state; (2) The identity connections improve the flow of information forward and backward through the network.

## Acknowledgments

We are grateful to Hugo Larochelle, Jan Chorowski, and students at MILA, especially Çağlar Gülçehre, Marcin Moczulski, Chiheb Trabelsi, and Christopher Beckham, for helpful feedback and discussions. We thank the developers of Theano (Theano Development Team, 2016), Fuel, and Blocks (van Merriënboer et al., 2015). We acknowledge the computing resources provided by ComputeCanada and CalculQuebec. We also thank IBM and Samsung for their support. We would also like to acknowledge the work of Pranav Shyam on learning RNN hierarchies. This research was developed with funding from the Defense Advanced Research Projects Agency (DARPA) and the Air

Force Research Laboratory (AFRL). The views, opinions and/or findings expressed are those of the authors and should not be interpreted as representing the official views or policies of the Department of Defense or the U.S. Government.

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

## 6 APPENDIX

### 6.1 STATIC IDENTITY CONNECTIONS EXPERIMENT

This experiment was suggested by AnonReviewer2 during the ICLR review process with the goal of disentangling the effects zoneout has (1) through noise injection in the training process and (2) through identity connections. Based on these results, we observe that noise injection is essential for obtaining the regularization benefits of zoneout.

In this experiment, one zoneout mask is sampled at the beginning of training, and used for all examples. This means the identity connections introduced are static across training examples (but still different for each timestep). Using static identity connections resulted in slightly lower *training* (but not validation) error than zoneout, but worse performance than an unregularized LSTM on both train and validation sets, as shown in Figure 7.

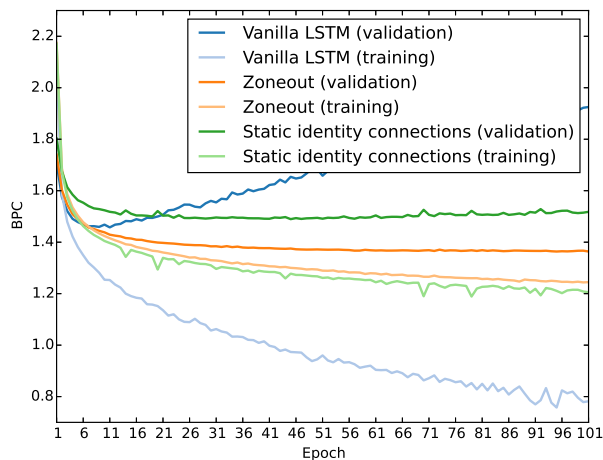

Figure 7: Training and validation curves for an LSTM with static identity connections compared to zoneout (both $Z_c = 0.5$ and $Z_h = 0.05$) and compared to a vanilla LSTM, showing that static identity connections fail to capture the benefits of zoneout.

