# Peer review of "Zoneout: Regularizing RNNs by Randomly Preserving Hidden Activations"

_ICLR 2017 — accepted_

[Official Review · AnonReviewer3 · rating 8 · confidence 5 · 17 Dec 2016]
originality 2 · clarity 2 · impact 2 · substance 2 · meaningful comparison 1

This paper tests zoneout against a variety of datasets - character level, word level, and pMNIST classification - showing applicability in a wide range of scenarios. While zoneout acts as a regularizer to prevent overfitting, it also has similarities to residual connections. The continued analysis of this aspect, including analyzing how the gradient flow improves the given tasks, is of great interest and helps show it as an inherent property of zoneout.

This is a well written paper with a variety of experiments that support the claims. I have also previously used this technique in a recurrent setting and am confident on the positive impact it can have upon tasks. This is likely to become a standard technique used within RNNs across various frameworks.

[Official Review · AnonReviewer2 · rating 8 · confidence 4 · 18 Dec 2016]
**Simple idea, well executed.**
originality 1 · clarity 5 · impact 2 · substance 2 · meaningful comparison 2

Paper Summary
This paper proposes a variant of dropout, applicable to RNNs, in which the state
of a unit is randomly retained, as opposed to being set to zero. This provides
noise which gives the regularization effect, but also prevents loss of
information over time, in fact making it easier to send gradients back because
they can flow right through the identity connections without attenuation.
Experiments show that this model works quite well. It is still worse that
variational dropout on Penn Tree bank language modeling task, but given the
simplicity of the idea it is likely to become widely useful.

Strengths
- Simple idea that works well.
- Detailed experiments help understand the effects of the zoneout probabilities
  and validate its applicability to different tasks/domains.

Weaknesses
- Does not beat variational dropout (but maybe better hyper-parameter tuning
  will help).

Quality
The experimental design and writeup is high quality.

Clarity
The paper clear and well written, experimental details seem adequate.

Originality
The proposed idea is novel.

Significance
This paper will be of interest to anyone working with RNNs (which is a large
group of people!).

Minor suggestion-
- As the authors mention - Zoneout has two things working for it - the noise and
  the ability to pass gradients back without decay. It might help to tease apart
the contribution from these two factors. For example, if we use a fixed
mask over the unrolled network (different at each time step) instead of resampling
it again for every training case, it would tell us how much help comes from the
identity connections alone.

[Official Review · AnonReviewer1 · rating 7 · confidence 4 · 20 Dec 2016 (modified: 12 Jan 2017)]
**Incremental improvement, not convincing enough**
soundness 2 · originality 4 · impact 2 · substance 2 · recommendation (unofficial) 1

The authors propose a conceptually simple method for regularisation of recurrent neural networks. The idea is related to dropout, but instead of zeroing out units, they are instead set to their respective values at the preceding time step element-wise with a certain probability.

Overall, the paper is well written. The method is clearly represented up to issues raised by reviewers during the pre-review question phase. The related work is complete and probably the best currently available on the matter of regularising RNNs.

The experimental section focuses on comparing the method with the current SOTA on a set of NLP benchmarks and a synthetic problem. All of the experiments focus on sequences over discrete values. An additional experiment also shows that the sequential Jacobian is far higher for long-term dependencies than in the dropout case.

Overall, the paper bears great potential. However, I do see some points.

1) As raised during the pre-review questions, I would like to see the results of experiments that feature a complete hyper parameter search. I.e. a proper model selection process,as it should be standard in the community. I do not see why this was not done, especially as the author count seems to indicate that the necessary resources are available.

I want to repeat at this point that Table 2 of the paper shows that validation error is not a reliable estimator for testing error in the respective data set. Thus, overfitting the model selection process is a serious concern here.
Zoneout does not seem to improve that much in the other tasks.

2) Zoneout is not investigated well mathematically. E.g. an analysis of the of the form of gradients from unit K at time step T to unit K’ at time step T-R would have been interesting, especially as these are not necessarily non-zero for dropout. Also, the question whether zoneout has a variational interpretation in the spirit of Yarin Gal’s work is an obvious one. I can see that it is if we treat zoneout in a resnet framework and dropout on the incremental parts. Overall, little effort is done answering the question *why* zoneout works well, even though the literature bears plenty of starting points for such analysis.

3) The data sets used are only symbolic. It would have been great if more ground was covered, i.e. continuous data such as from dynamical systems. To me it is not obvious whether it will transfer right away.


An extreme amount of “tricks” is being published currently for improved RNN training. How does zoneout stand out? It is a nice idea, and simple to implement. However, the paper under delivers: the experiments do not convince me (see 1) and 3)). There authors do not provide convincing theoretical insights either. (2)

Consequently, the paper reduces to a “epsilon improvement, great text, mediocre experimental evaluation, little theoretical insight”.

[Final Decision · Program Chairs · 06 Feb 2017]
**ICLR committee final decision**

Very nice paper, with simple, intuitive idea that works quite well, solving the problem of how to do recurrent dropout.
 
 Pros:
 - Improved results
 - Very simple method
 
 Cons:
 - Almost the best results (aside from Variational Dropout)